

# Curd, seed yield and disease resistance of cauliflower are enhanced by oligosaccharides

Md. Mijanur Rahman Rajib[1,2,3,*], Hasina Sultana[3,*], Jin Gao[1], Wenxia Wang[1] and Heng Yin[1,2]

[1] Natural Products and Glyco-Biotechnology Lab, Dalian Institute of Chemical Physics, Chinese Academy of Sciences, Dalian, Liaoning, China
[2] University of Chinese Academy of Sciences, Beijing, China
[3] Department of Horticulture, Bangabandhu Sheikh Mujibur Rahman Agricultural University, Gazipur, Bangladesh
* These authors contributed equally to this work.

Corresponding authors
Md. Mijanur Rahman Rajib,
mmrrajib@bsmrau.edu.bd
Jin Gao, gaojin0903@dicp.ac.cn

## ABSTRACT

**Background:** Oligosaccharides have been demonstrated as promoters for enhancing plant growth across several crops by elevating their secondary metabolites. However, the exploration of employing diverse oligosaccharides for qualitative trait improvements in cauliflower largely unknown. This study was intended to uncover the unexplored potential, evaluating the stimulatory effects of three oligosaccharides on cauliflower's curd and seed production.

**Methods:** Two experiments were initiated in the early (15 September) and mid-season (15 October). Four treatments were implemented, encompassing a control (water) alongside chitosan oligosaccharide (COS 50 mg.L$^{-1}$) with a degree of polymerization (DP) 2–10, oligo galacturonic acid (OGA 50 mg.L$^{-1}$) with DP 2–10 and alginate oligosaccharide (AOS 50 mg.L$^{-1}$) with DP 2–7.

**Results:** Oligosaccharides accelerated plant height (4–17.6%), leaf number (17–43%), curd (5–14.55%), and seed yield (17.8–64.5%) in both early and mid-season compared to control. These enhancements were even more pronounced in the mid-season (7.6–17.6%, 21.37–43%, 7.27–14.55%, 25.89–64.5%) than in the early season. Additionally, three oligosaccharides demonstrated significant disease resistance against black rot in both seasons, outperforming the control. As a surprise, the early season experienced better growth parameters than the mid-season. However, performance patterns remained more or less consistent in both seasons under the same treatments. COS and OGA promoted plant biomass and curd yield by promoting Soil Plant Analysis Development (SPAD) value and phenol content. Meanwhile, AOS increased seed yield (56.8–64.5%) and elevated levels of chlorophyll, ascorbic acid, flavonoids, while decreasing levels of hydrogen per oxide (H$_2$O$_2$), malondialdehyde (MDA), half maximal inhibitory concentration (IC$_{50}$), and disease index. The correlation matrix and principal component analysis (PCA) supported these relations and findings. Therefore, COS and OGA could be suggested for curd production and AOS for seed production in the early season, offering resistance to both biotic and abiotic stresses for cauliflower cultivation under field conditions.

# INTRODUCTION

Cauliflower is a cruciferous vegetable of the esteemed family Brassicaceae (*Nimkar & Korla, 2014*). Beyond its culinary appeal, cauliflower serves as a vital source of precious dietary fiber, a nutritive composition, and valuable bioactive compounds globally. The nutritional composition of cauliflower includes notable amounts of protein (1.92 g), fat (0.28 g), carbohydrate (4.97 g), and fiber (2 g). With a lineup that includes essential nutrients such as Ca (22 mg), Fe (0.42 mg), Mg (15 mg), P (44 mg), K (299 mg), Na (30 mg), Zn (0.27 mg), Cu (0.039 mg), Mn (0.155 mg), Se (0.6 μg), vitamin C (48.2 mg), folate (57 μg), vitamin E (0.08 mg), vitamin K (15.5 μg), lutein + zeaxanthin (1 μg), and glucosinolate (1,178 mg) (*Agagunduz et al., 2022*). The symphony of phytochemicals creates the flavor, color, and aroma of cauliflower. Furthermore, cruciferous vegetables contain a wide array of bioactive compounds that deliver a spectrum of health benefits to humanity (*Acimovic et al., 2016*; *Raiola et al., 2018*). These bioactive compounds consist of glucosinolates, S-methyl cysteine sulfoxide, secondary metabolites (flavonoids and non-flavonoids), phenolic compounds, and dietary fiber. These compounds are responsible for the inactivation and inhibition of carcinogenic and tumor cells, exhibiting antiviral and antibacterial properties. These compounds are responsible as the guardians of antioxidant capacity, forming a fortress against oxidative threats (*Fusari et al., 2020*). Correlative inhibitory effects on reducing *Xanthomonas campestris* pv. *Campestris* (black rot) virulence in the Brassicaceae have been reported for glucosinolates, phenolics, and antioxidant capacity (*Aires et al., 2011*; *Baranek et al., 2021*). However, white curds of cauliflower often suffer uneven purple pigmentation due to anthocyanin biosynthesis influenced by environmental factors, adversely impacting both appearance and economic value (*Chen et al., 2022*). Secondary metabolites of flavonoidpigments attract pollinators and seed carriers (*Hatier & Gould, 2008*; *Yuan, Byers & Bradshaw, 2013*), boosting plant immunity to diverse stresses (*Gould, 2004*).

In 2020, global cauliflower production reached a substantial 25.50 million metric tons (*Prodhan et al., 2022*), while the demand is always intensifying for the mountain population worldwide. Asia dominates cauliflower production, with China and India contributing to 75% of the world's cauliflower. Despite this, challenges persist in achieving satisfactory production and quality. These challenges are primarily related to planting materials, variety and seed selection, cultural management, climatic conditions, and edaphic factors. To diminish these issues, the extensive use of agrochemicals exacerbates severe global crises concerning soil, the environment, and public health. In pursuing sustainable crop production, there is an obvious need for an efficient and eco-friendly alternative (*Ahmed et al., 2020*). Oligosaccharides (OSs) emerge as a promising solution, promoting morphological, phenological, physiological, and biochemical attributes while elevating secondary metabolites and bioactive compounds in sustainable agriculture. Their

positive impact extends to nutrient efficacy, soil health, and immunity against biotic and abiotic stress (*He et al., 2021*).

The edible part of cauliflower, called curd, is botanically the pre-condition of inflorescence. Cauliflower life span comprises three phases namely growth, curd, and flowering as well as the seed phase (*Kaur & Mal, 2018*). High and low temperatures during the growth phase led to premature buttoning, resulting in blind buds or no head formation. Naturally available non-hazardous oligosaccharides such as chitosan (COS), alginate (AOS), pectin (POS), xyloglucan (XOS) and oligo-galacturonic acid (OGA) have been focused so far to minimise biotic and abiotic stresses (*He et al., 2021*; *Wan et al., 2021*; *Ahmed et al., 2020*; *Jia, Rajib & Yin, 2020*; *Zhang et al., 2020*). Studies have demonstrated enhancements in nitrogen use efficacy, soil ecosystem, plant growth, and physiology by applying these OSs. Chitosan, oligo-chitosan, and oligo-alginate defend against viruses, bacteria, and fungi by activating SA or JA/ET signaling pathways, and orchestrating the accumulation of secondary metabolites and phenolic compounds. OSs exhibit efficacy against drought, salinity, chilling, and heavy metal stresses, forming a shield through an ABA-dependent pathway. OSs also enhance plant growth through JA/ET signaling pathways, increasing photosynthesis, hormone contents (auxin and gibberellins), and C-N assimilation (*Moenne & Gonzalez, 2021*).

Different forms of COS promote growth by influencing related attributes in various plants, including grapevine (*Barka et al., 2004*), orchid (*Nge et al., 2006*), bean (*Chatelain, Pintado & Vasconcelos, 2014*; *Vasconcelos, 2014*), tomato (*Sathiyabama, Akila & Charles, 2014*), potato (*Harshouf et al., 2017*), and rape (*Yin et al., 2006*). It also enhanced seedlings' growth and fresh biomass by photosynthesis, resulting in 50% acceleration (*Zhang, Zheng & Zhang, 2016*), and carbon-nitrogen (C-N) assimilation (*Zhang et al., 2017*). COS also protected pear fruits from *Alternaria kikuchiana* (*Meng et al., 2010*) and rape from *Sclerotinia sclerotiorum* (*Yin et al., 2013*) either inducing SA or JA/ET pathway (*Jia et al., 2020*). Additionally, COS also generated signaling molecules (Ca$^{2+}$, H$_2$O$_2$, NO), enzymes, and phenolic compounds (anthocyanins) as antimicrobial and abiotic properties in grapevine (*Singh et al., 2020*). AOS, with different sources, concentrations, and degrees of polymerization (DP, 2–8) influences plant growth and development. AOS increased root number in barley (*Natsume et al., 1994*) and lettuce (*Iwasaki & Matsubara, 2000*) while root length and fresh biomass were improved in rice, peanut (*Hien et al., 2000*; *Zhang et al., 2014*), *Limonium*, *Lisianthus*, chrysanthemum (*Luan et al., 2003*), barley, bean (*Luan et al., 2009*), and maize (*Hu et al., 2004*). AOS impacted shoot/leaf biomass, photosynthesis/chlorophyll content, and C-N assimilation in *Artemisia annua* (*Aftab et al., 2011*), poppy (*Khan et al., 2011*), and mint (*Idrees et al., 2011*). AOS also exhibited antipathogenic activities in rice (*Zhang et al., 2016*), soybean (*Peng et al., 2018*), and *Arabidopsis* (*Zhang et al., 2019*) *via* the SA-dependent pathway. Similarly, OGA acted as a growth inhibitor of etiolated tomato seedlings (*Zhou et al., 2023*) and a ripening promoter of tomato fruits (*Ma et al., 2016*). OGA demonstrated similar antipathogenic activity in grapes, tobacco, and *Arabidopsis* (*Ferrari et al., 2008*; *Howlader et al., 2020*) through the SA/JA pathway.

To date, OS research has focused on invigorating growth, improving yield attributes, and increasing innate immunity in plants. Limited research has explored the morpho-

physiological, phenological intricacies, quality, and yield of curd, seed production, and the immune system of cauliflower post-treatment with various OSs. Therefore, out of the OSs under review, COS, AOS, and OGA are considered for comparison in terms of quality production and immune resistance of cauliflower. Two experiments in early (late autumn) and mid-season (early winter) at the "Horticultural Research Field" of Bangabandhu Sheikh Mujibur Rahman Agricultural University, Bangladesh, were set up with the newly released seed-producing variety BU cauliflower 1 (BU1). This study was meticulously designed to scrutinize the impact of diverse OSs (COS, AOS, and OGA) on cauliflower's biological, physiological, phenological, and biochemical parameters. Concurrently, it aimed to unravel the complex relationship between OSs and cauliflower developmental or immune traits. Moreover, this study is expected to provide convincing evidence that COS, AOS, and OGA can strengthen traits related to yield, environmental protection, and safety while enhancing disease resistance through the promotion of secondary metabolites and linked compound production.

# MATERIALS AND METHODS

## Study location

Bangabandhu Sheikh Mujibur Rahman Agricultural University (BSMRAU), Bangladesh, served as the experimental site. The experimental area, located within the Salna Series of Madhupur Tract, featured clay loam soil with shallow Red Brown Terrace type soils falling under Agroecological Zone 28 (AEZ 28) at coordinates 24°23′N and 90°08′E (*Prodhan et al., 2022*). The soil conditions were slightly acidic (pH 6.2), low in organic matter (0.86%), total N of 0.09%, available P of 10.22 microgram.$g^{-1}$, and exchangeable K of 0.07 milliequivalents.100 $g^{-1}$ (Table S1). The subtropical zone experiences variations in temperatures during summer (maximum 32 °C, minimum 27 °C, and average 29 °C) and winter (maximum 25 °C, minimum 15 °C, and average 19.6 °C). The geographical context is provided in Fig. S1 and corresponding weather conditions are outlined in Table S2.

## Seeding and seedling

Cauliflower seeds (BU cauliflower-1) sourced from the Department of Horticulture, BSMRAU, Bangladesh, were employed for the study. Autoclaved pulverized soil was evenly distributed in a plastic tray to facilitate dense sowing, with subsequent prickling performed after 10 days. Polybags were filled with an equivalent blend of autoclaved soil and putrefied cowdung mixture as the growth medium. Following stabilization of the seedlings, they were placed within a polytunnel for 20 days, maintaining a temperature of 28 ± 2 °C and a light/dark cycle of 12 h.

## Experimental design, layout, and transplanting of seedling cultural practices and aftercare

Two distinct experiments were laid out in a single-factor randomized complete block design with three replications. Three different oligosaccharides along with control were considered as treatments (Control (No OS); COS @ 50 mg.$L^{-1}$, OGA @ 50 mg.$L^{-1}$ and AOS @ 50 mg.$L^{-1}$). Each treatment was assigned three plots measuring 0.12 m × 0.12 m,

with interplot distances of 0.5 m. Within each treatment, 12 plants were distributed across three plots, 12 plots in total for the four treatments. The three blocks were spatially separated by 1 m. Disease-free healthy seedlings were transplanted in the rows of 60 cm × 60 cm, following meticulous removal from the polybag. Thirty-day-old seedlings were transplanted in two consecutive field trials on 15 September and 15 October. All other cultural management was maintained except pest management, as recommended (*Azad et al., 2017*).

## Source of oligosaccharides and application

All the oligosaccharides (OSs) were provided by Dalian GlycoBio Co. Ltd. (Dalian, China). Chitosan oligosaccharide (COS) and oligo-galacturonides (OGA) were procured with DP 2 to 10, while alginate oligosaccharide (AOS) was DP 2 to 7 (*Zhang et al., 2019*; *Bose et al., 2019*; *Howlader et al., 2020*). The 1,000 mg.L$^{-1}$ stock solution was prepared by dissolving 1.0 g of OS in 1 L of distilled water and then stored at 4 °C. Prior to field application, the respective OS solution was diluted to 50 mg.L$^{-1}$ and sprayed at 15, 30, 45, and 60 days after transplanting (DAT). Additionally, gibberellic acid (GA3) at a concentration of 50 mg.L$^{-1}$ was applied at 25 and 40 DAT. GA3 was initially dissolved in 1 M sodium hydroxide (NaOH), following the same protocol as the OS preparation.

## Data collection

From each experimental unit, comprising 12 plants per treatment, five plants were randomly selected for comprehensive examination of various characteristics. The morphological parameters assessed included plant height at 75 days after transplanting (DAT), number of leaves at curd initiation, length of the biggest leaf at 75 DAT, the breath of the biggest leaf at 75 DAT, curd length, curd diameter, biological yield, marketable yield, curd yield, the height of seed stalk, number of primary, secondary and tertiary flowering branch per plant, number of secondary flowering branch per plant, number of pods per plant, length, and diameter of pod, number of seeds per pod, weight of seeds per plant (weight basis). Of the 12 randomly chosen plants, five were specifically selected for the collection of morphological and yield-related data, while an additional five plants were randomly chosen from the remaining seven for the evaluation of seed yield attributed parameters. The seed stalks were meticulously cut and dried under shade conditions when 75% of the pods exhibited the transition from green to yellow and brown.

## Disease index

To assess the disease index and pathogenic disease symptoms associated with *Xanthomonas campestris* pv. *campestris* (Black rot) before harvesting cauliflower curd at 75 DAT, the following method was used (*Zhang et al., 2019*): first, we counted how many leaves had necrotic lesion areas on each leaf (the proportion (S) of necrotic lesion areas) for each disease level: level 1 ($0 < S \le 0.25$); level 2 ($0.25 < S \le 0.5$); level 3 ($0.5 < S \le 0.75$) and level 4 ($0.75 < S \le 1$). The following formula was applied to obtain the disease index.

$$Disease\,index\,(\%) = (\Sigma\,level \times leaves\,per\,level/total\,leaves \times the\,highest\,level) \times 100 \quad (1)$$

## Measure SPAD value and estimation of chlorophyll (mg.g$^{-1}$ FW)

The Soil Plant Analysis Development (SPAD) value indicates the proportional chlorophyll content within the leaf. The portable SPAD meter is used for swift and precise chlorophyll detection in agriculture (*Ling, Huang & Jarvis, 2011*). SPAD values were recorded using the same plants' leaves twice. Once was before 4th spraying, and once was 5 days after 4th spraying of oligosaccharides by measuring with a SPAD-502 Plus chlorophyll meter (Konica-Minolta, Tokyo, Japan). Then, the changes between earlier and later values were expressed in percentages (%). Fresh leaves were then collected for chlorophyll content in the laboratory.

Using a double beam spectrophotometer (model APEL, UV-VIS Spectrophotometer, PD 303 UV, PD 33-3-OMS-101 b, Kawaguchi, Japan), chlorophyll content was determined on a fresh weight basis of the leaf samples with 80% acetone. *Witham, Blades & Devin (1986)* proposed the following equations for estimating different chlorophylls. Cauliflower leaves were measured at 5 mg. The sample was dipped in 80% acetone in a test tube, and the volume was made up to 25 mL. We kept the sample in a dark place for 48 h in a test tube covered with aluminium foil. A spectrophotometer measured the absorbance of 663 and 645 nm of the filtrate. The chlorophyll content of the sample was calculated by the following formula:

$$Chlorophyll\,a\,(mg.g^{-1}) = \frac{[12.7 \times (OD663) - 2.69(OD645)] \times V}{W \times 1{,}000} \quad (2)$$

$$Chlorophyll\,b\,(mg.g^{-1}) = \frac{[22.9 \times (OD645) - 4.68(OD663)] \times V}{W \times 1{,}000} \quad (3)$$

$$Chlorophyll\,total\,(mg.g^{-1}) = \frac{[20.2 \times (OD645) + 8.02(OD663)] \times V}{W \times 1{,}000} \quad (4)$$

where, $OD_{645}$ = Optical density at 645 nm wavelength, $OD_{663}$ = Optical density at 663 nm wavelength, V = Volume of the extract, W = Fresh weight in grams of the tissue extracted.

## Analysis of total phenolic, total flavonoids and antioxidant activity

An extraction of 1 g powder of freshly blended curd sub-sample with 25 mL methanol was used to determine the curd's total phenolic content and antioxidant activity. After 2.5 h of incubation at 30 °C, the sample was centrifuged for 15 min at 6,000 rpm. After decanting and filtering the supernatants using Whatman 42 filter paper, the supernatants were stored at 4 °C until analysis was performed.

Based on the Folin-Ciocalteu method (*Singleton & Rossi, 1965*) with some modifications, the total phenolic content (TPC) of the samples was determined. The sample extracts were shaken with 2.5 mL of Folin–Ciocalteu reagent in the presence of 0.5 mL of Folin–Ciocalteu reagent for 10 min before analysis. Next, the mixture was incubated at 30 °C for 1 h after adding 2 mL of 7.5% sodium carbonate to the solution. A known

concentration of gallic acid (polyphenolic substance) was used to standardize the absorbance of diluted extracts (10, 20, 40, 60, 80, 100, and 200 μL). A UV-VIS (PD-303 UV Spectrophotometer; APEL Co., Kawaguchi, Japan) spectrophotometer was used to measure the absorbance of the sample and the gallic acid standard at 760 nm. TPC was expressed in mg of gallic acid equivalents per 100 g of fresh weight.

A colorimetric technique with aluminium chloride was used to determine the total flavonoids content (*Nyangena et al., 2019*). Working samples were derived from methanolic samples previously used to determine total phenolic content. In an Eppendorf tube, 100 μL of diluted extract solution was added to 400 μL of methanol. Initially, 100 μL of 10 % aluminium chloride hexahydrate ($AlCl_3 \cdot 6H_2O$) was added, followed by 100 μL of 1 M sodium acetate. An absorbance reading at 420 nm was performed on the reaction mixture after 40 min of incubation. TPCs were expressed in mg of quercetin equivalents per 100 g (fresh weight).

DPPH (2,2-diphenyl-1-picrylhydrazyl) radical scavenging assay was used to measure fresh cauliflower curd antioxidant activity. Antioxidants are tested in this assay for their ability to scavenge stable radicals. The study was conducted according to *Nyangena et al. (2019)* with some modifications. Previously prepared extracts were used to assess antioxidant activity. To perform an antioxidant assay, methanol up to 3 mL was added to extracts and ascorbic acid solutions at 20, 40, 80, 100, and 200 $\mu g.mL^{-1}$ concentrations. A methanolic DPPH solution of 1 mL was added (0.004 mg DPPH added to 100 mL of methanol). A spectrophotometer was used to measure the reading at 517 nm against a blank (control) after 30 min of keeping the reaction mixture in a dark place. According to the following formula (*Sridhar & Charles, 2019*), the radical scavenging activity was estimated ($\mu g.mL^{-1}$) on a dry basis as follows:

$$\% \text{ scavenging activity} = \frac{(A0 - A1)}{A0} \times 1{,}000 \qquad (5)$$

where, A0 = Absorbance of control and A1 = Absorbance of sample

Based on the graph that plotted the percent radical scavenging activity against the concentration of extract for standards and test samples, inhibition concentration ($IC_{50}$) is used to determine antioxidant capacity. Higher antioxidant activity is associated with a lower $IC_{50}$ value (*Dhanani et al., 2017*). Below is the formula used to calculate $IC_{50}$.

$$IC_{50} = \frac{(y - b)}{a} \qquad (6)$$

where the above equation substitutes 50 for y; a and b are determined by plotting regression lines separately for each sample.

## Determination of ascorbic acid (mg.100 $g^{-1}$ FW)

Fresh curd was tested for its ascorbic acid content using a titration method with some modifications (*Elgailani et al., 2017*). A 50 mL conical flask was filled with 5 mL of extract solution prepared from 1 g of blended sample powder. The extract solutions were treated with 5.0 mL of potassium iodide (KI) solution (5%) and 2 mL of glacial acetic acid,

followed by 2 mL of starch solution (2%) and 5 mL of KI solution (5%). Titrations were conducted from the burette against 0.001 N potassium iodate ($KIO_3$) until the solution turned blue.

Using the results, we calculated the ascorbic acid concentration per sample (mg.100g$^{-1}$).

$$\text{Ascorbic acid content mg.100g}^{-1} = \frac{(T \times F \times Vt \times 100)}{(v \times W)} \tag{7}$$

where, T = titrated volume of 0.001N $KIO_3$ (mL), F = 0.088 mg of ascorbic acid per mL of 0.001 N KIO3, Vt = total volume of sample extracted (mL), v = volume of the extract (mL) titrated with 0.001 N KIO3.

## Determination of hydrogen peroxide ($H_2O_2$)

$H_2O_2$ concentration in the tissue was determined by spectrophotometry (APEL, UV-VIS Spectrophotometer, PD 303 UV, PD 33-3-OMS-101 b, Kawaguchi, Japan) as mentioned by *Loreto & Velikova (2001)*. After 72 h of 3$^{rd}$ OS application, a fresh sample weighing 1 g was homogenized with three replications for each treatment in 3 mL of trichloroacetic acid (TCA) (1%) and centrifuged for 10 min at 4 °C with 10,000 × g. After centrifugation, 0.75 mL of the supernatant was added to 0.75 mL of 10 mM potassium phosphate buffer (pH 7.0) containing 1.5 mL of 1 M KI. Based on a molar extinction coefficient of 0.28 mol.cm$^{-1}$, the $H_2O_2$ content was calculated and expressed as mg.g$^{-1}$ FW.

$$H_2O_2 = \frac{(A390 - ABlank)}{Co-efficient} \times \text{Sample amount} \times \text{Buffer volume.} \tag{8}$$

## Determination of malondialdehyde

Malondialdehyde (MDA) levels were used to measure lipid peroxidation. Initially, 20 g of trichloroacetic acid (TCA) granules and 80 mL of distilled water were combined to generate a 20% (w/v) trichloroacetic acid (TCA) stock solution, and then the volume was up to 100 mL of distilled water. After that, 100 mL of distilled water was added to 5 g of TCA granules to make an extraction buffer (5% (w/v) TCA solution). The solution was stored at 4 °C. In a subsequent step, a solution of TBA 20% (w/v), TCA 0.35 g was prepared in 70 mL to prepare the TBA reaction mixture (TBA reagent). In 3 mL of a 5% TCA solution, fresh leaves (0.5 g) were homogenized. Using 15,500 g at 4 °C, the homogenate was centrifuged for 15 min. Afterward, 1 mL of the supernatant was added to 4 mL of the reaction mixture, which was heated to 95 °C for 30 min. A second centrifugation of 15,500 g for 10 min was performed after the solution cooled. At 532 and 600 nm, colored supernatants were measured. To calculate the MDA content on a fresh weight basis, the following formula was used:

$$MDA(nmol.g^{-1}FW) = (A532 - A600)E \times 1,000 \times V \times W \tag{9}$$

where, A532 and A600 was reading at 532 and 600 nm; E was the extinction coefficient (Lambert-Beer law) = 155 mM.cm$^{-1}$. V= volume of the reaction mixture (supernatants) and W = fresh weight of the initial sample.

## Statistical analyses

A randomized complete block design (RCBD) was employed to design two independent experiments with three replications, based on a single factor. Within each replication, ten plants out of twelve were systematically chosen for data collection in each treatment. Microsoft Excel facilitated the computation of average values from tabulated data, which were subsequently used as experiment data. R (version 4.1.2) and SPSS (version 22) were used to analyze the experiment data. An analysis of variance (ANOVA) was performed to identify significant differences between samples following a Tukey HSD test at a 5% level of significance. The interrelationships among crucial growth and yield attributes were elucidated through correlation matrices and cluster analysis. Subsequently, principal component analysis (PCA) was executed to visually represent correlated cauliflower attributes, denoted Dim1 (Dimension1 or PC1), Dim2 (Dimension2 or PC2). Variables were selected based on the eigenvalues of the principal components, which indicate the total amount of variance each principal component can explain. This approach focused on the factor loadings of the independent variables (oligosaccharides) and the contributions of each dependent variable (attributes) to the total variation for any component (factor) with an eigenvalue > 1.00. Various R packages (agricultural, facatominer, factoextra, ggplot2, corrplot) were used to determine the factor loadings and contributions (4.1.2). Statistical analysis was performed using means of three determinations with standard deviations.

# RESULTS

## BU cauliflower 1 morphology as influenced by oligosaccharides

The influence of diverse OSs on plant growth characteristics was evaluated 75 days after transplanting (DAT), encompassing parameters such as plant height, leaf number, length, and breadth of largest leaves (Table 1). A significant variation was apparent among the treatments in both experiments. Plants treated with OSs consistently displayed elevated plant height compared to the control, with a marginally more efficient increase in mid-season than in the early season. The increment in plant height ranged from 4–7% in the early season and escalated to 7.6–17.6% in the mid-season experiment. Remarkably, COS performed better among the OSs, followed by OGA and AOS in both experiments. The tallest plant (50.06 cm) was detected in the OGA-treated plot during the early season, while COS produced the tallest plant (48.99 cm) in the mid-season. OGA, COS, and AOS demonstrated identical performance in the early season but significantly diverged in the mid-season.

The number of leaves per plant reflected plant height patterns. Leaf number increased by 17–43% compared to control. This increment was up to 17–32% in the early season and 21.37–43% in the mid-season experiment. OSs induced a remarkable increase in a number of leaves to control. OGA expressed the maximum leaves number (28.67) in the early season, while COS (27.55) led in the mid-season. Leaf numbers from OGA and COS were identical but differed significantly from AOS in both seasons. Scientific literature indicates a positive correlation between cauliflower curd initiation and leaf number. Curd initiation highly relies on the number of leaves and temperature during vegetative growth (*Booij &*

**Table 1 Morphological attributes of BU cauliflower 1 under different oligosaccharides treatments.**

| Treatment[X] | Early season (15 September 2021)[Y] | | | | Mid-season (15 October 2021) | | | |
|---|---|---|---|---|---|---|---|---|
| | Plant height (cm) | Leaf number | Leaf length (cm) | Leaf breadth (cm) | Plant height (cm) | Leaf number | Leaf length (cm) | Leaf breadth (cm) |
| Control | 46.78 ± 0.75 | 21.67 ± 0.72 | 36.28 ± 0.42 | 20.67 ± 0.66 | 41.63 ± 0.57 | 19.28 ± 0.65 | 34.83 ± 0.64 | 19.84 ± 0.84 |
| COS 50 | 50.00 ± 1.42 | 28.11 ± 1.89 | 39.39 ± 0.19 | 22.50 ± 1.19 | 48.99 ± 1.18 | 27.55 ± 1.95 | 38.87 ± 0.78 | 22.06 ± 1.33 |
| OGA 50 | 50.06 ± 0.82 | 28.67 ± 0.33 | 38.72 ± 1.08 | 22.72 ± 0.75 | 47.39 ± 0.93 | 27.23 ± 0.52 | 37.56 ± 1.27 | 22.42 ± 0.85 |
| AOS 50 | 48.72 ± 0.97 | 25.45 ± 1.07 | 38.89 ± 0.67 | 21.94 ± 0.63 | 44.83 ± 1.09 | 23.40 ± 0.80 | 37.98 ± 0.88 | 21.21 ± 0.48 |
| Level of significant | *** | *** | ** | ** | *** | *** | ** | ** |
| LSD (0.05%) | 1.37 | 2.78 | 1.91 | 1.41 | 0.91 | 1.77 | 1.90 | 1.13 |

Notes:
[X] All treatments were applied at 15, 30, 45 and 60 days after transplanting @ 50 mg.L$^{-1}$ except control (sterile water); COS, chitosan oligosaccharide; OGA, oligo galacturonic acid; AOS, alginate oligo saccharide; values are as the means ± SD in two independent measurements; LSD, least significance difference value at 0.05%; asterisks indicate significant differences (**$P < 0.01$, ***$P < 0.001$).
[Y] cm, centimetre.

Struik, 1990; Wurr & Fellows, 2000). Lower temperatures (5 °C) result in earlier curd initiation with fewer leaves, while higher temperatures (20 °C) necessitate more leaves for curd initiation (Williams & Atherton, 1990). The plant experienced temperatures above 20 °C for 3 months in the early season and 2 months in the mid-season during the vegetative growth phase (Table S2). In both conditions, OSs produced more leaves compared to control, indicating their proficiency in accelerating biomass production.

Leaves function as conduits for transferring photosynthetic products from the source (leaf) to the sink (curd, flower, pod, seed). Longer and wider leaves contain more surface area for photosynthetic activity and contribute to curd and seed yield. Leaf length and breadth appeared significantly extended and expanded compared to the control. COS outperformed other OSs, providing the longest leaf (39.39 and 38.87 cm) in early and mid-season experiments, respectively. On the contrary, OGA exhibited the broadest leaves (22.72 and 22.42 cm) in the early and mid-season (Table 1). However, AOS had shorter leaf length (38.89 and 37.98 cm) and breadth (21.94 and 21.21 cm) in both early and mid-season. Despite this, AOS exhibited identical behavior to other OSs and significantly differed from the control. These findings emphasize OSs' ability to stimulate biomass production under fluctuating field conditions.

## Curd yield and attributes as influenced by oligosaccharides

Curd length and diameter correlated proportionally with curd yield. Curd size, particularly extra-large (20–30 cm), significantly interfered with yield, yield attributes, and quality of cauliflower seeds (Ahmad et al., 2018). Although curd size improved with OSs treatments compared to the control, the improvement was not statistically significant, except for curd length in the early season. OGA performed identically better among the OSs in terms of curd length (19.56 and 19.35 cm) and diameter (20.56 and 20.35 cm) during both early and mid-season, respectively (Table 2).

The entire above-ground biomass was quantified as biological yield, while removing leaves up to the curd level constituted marketable yield. Curd devoid of leaves was

**Table 2 Yield attributes of BU cauliflower 1 under different oligosaccharides treatments.**

| Treatment[X] | Early season (15 September 2021)[Y] | | | | | Mid-season (15 October 2021) | | | | |
|---|---|---|---|---|---|---|---|---|---|---|
| | Curd length (cm) | Curd diameter (cm) | Biological yield (kg) | Marketable yield (kg) | Curd yield (kg) | Curd length (cm) | Curd diameter (cm) | Biological yield (kg) | Marketable yield (kg) | Curd yield (kg) |
| Control | 17.33 ± 0.33 | 18.33 ± 0.52 | 1.46 ± 0.07 | 1.00 ± 0.02 | 0.58 ± 0.02 | 16.53 ± 0.40 | 17.97 ± 0.51 | 1.42 ± 0.13 | 0.95 ± 0.12 | 0.55 ± 0.02 |
| COS 50 | 18.33 ± 1.20 | 19.33 ± 2.21 | 1.76 ± 0.15 | 1.11 ± 0.08 | 0.64 ± 0.06 | 18.21 ± 1.09 | 19.14 ± 1.18 | 1.75 ± 0.14 | 1.09 ± 0.08 | 0.63 ± 0.06 |
| OGA 50 | 19.56 ± 2.21 | 20.56 ± 1.20 | 1.76 ± 0.21 | 1.10 ± 0.25 | 0.61 ± 0.12 | 19.35 ± 2.05 | 20.35 ± 2.26 | 1.74 ± 0.22 | 1.06 ± 0.23 | 0.59 ± 0.11 |
| AOS 50 | 18.00 ± 1.00 | 19.00 ± 1.00 | 1.67 ± 0.22 | 1.07 ± 0.09 | 0.63 ± 0.08 | 17.64 ± 1.08 | 18.81 ± 0.91 | 1.62 ± 0.21 | 1.04 ± 0.09 | 0.61 ± 0.07 |
| Level of significant | ** | NS | ** | NS | ** | NS | NS | NS | NS | NS |
| LSD (0.05%) | 2.32 | – | 0.23 | – | 0.13 | – | – | – | – | – |

Notes:
[X]All treatments were applied at 15, 30, 45 and 60 days after transplanting @ 50 mg.L$^{-1}$ except control (sterile water); COS, chitosan oligosaccharide; OGA, oligo galacturonic acid; AOS, alginate oligo saccharide; values are as the means ± SD in two independent measurements; LSD, least significance difference value at 0.05%; NS, not significant; asterisks indicate significant differences (**$P < 0.01$).
[Y]cm, centimetre; kg, kilogram.

expressed as curd yield. All the yield-related parameters increased in the OSs-treated plants in comparison to the control at both seasons. The biological and curd yield were statistically significant only in the early season, with COS producing the maximum yield (1.76 and 0.64 kg, respectively) (Table 2). Curd yield was accelerated by about 5–14.55% compared to control, with a higher acceleration (7.27–14.55%) observed in mid-season compared to the early season (5–11%). Although COS performed comparatively better in biological and marketable yield at both seasons, it exhibited similar result to OGA and was followed by AOS treatment, which did not show significant differences between them (Fig. 1 and Table 2). A similar result was also observed in COS-treated plants concerning curd yield, where AOS accounted next to COS in both seasons.

## Phenological traits induced by oligosaccharides

Similar to plant height, stalk height increased in OSs treated plants compared to control in both seasons. Here, AOS showed the highest length (91.33 and 90.41 cm) in the early and mid-season. However, COS<OGA also performed statistically similar results to AOS concerning stalk height (Table 3 and Fig. 2).

The number of branches in the stalk was also measured. Although no significant difference was discerned in primary and secondary branches in both seasons, a conspicuous improvement trend was evident in OSs-treated plants compared to control. Only tertiary branches were recorded significantly more than control in the early season after being treated by OSs (Table 3). However, the better trend of primary (10.67 and 10.56), secondary (34.33 and 33.89), and tertiary branches (59 and 59.98) was noted in AOS treated plants in both seasons, notwithstanding the nonsignificant differences compared with control at primary and secondary branches. COS and OGA performed identical to AOS treated plants.

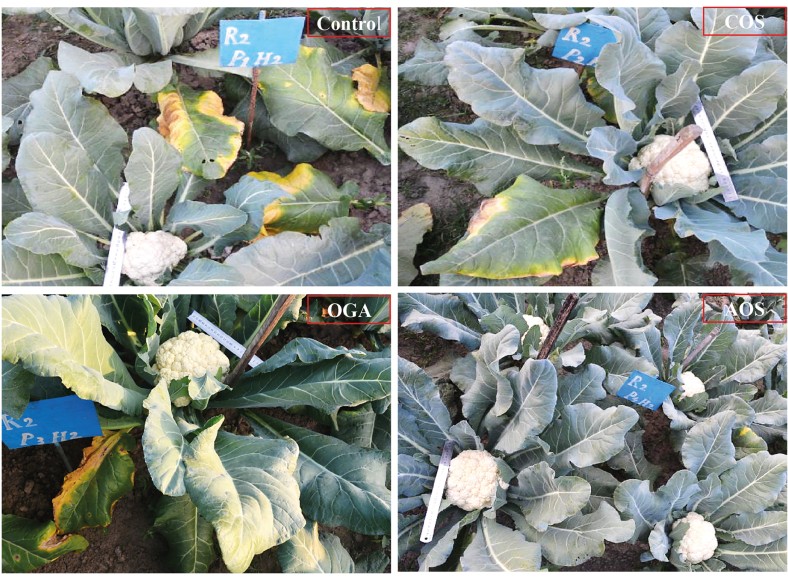

**Figure 1** **Biological yield, curd yield and black rot disease severity of BU cauliflower 1 as influenced by different oligosaccharides compare to control.** All treatments were applied at 15, 30, 45 and 60 days after transplanting @ 50 mg.L$^{-1}$ except control (sterile water); COS, chitosan oligosaccharide; OGA, oligo galacturonic aci; AOS, alginate oligosaccharide.

**Table 3** **Phenological attributes of BU cauliflower 1 under different oligosaccharides treatments.**

| Treatment[X] | Early season (15 September 2021)[Y] | | | | | Mid-season (15 October 2021) | | | | |
|---|---|---|---|---|---|---|---|---|---|---|
| | Stalk height (cm) | Primary branch | Secondary branch | Tertiary branch | Total pod | Stalk height (cm) | Primary branch | Sec Br | Ter Br | Total pod |
| Control | 77.00 ± 4.04 | 9.67 ± 1.52 | 31.33 ± 4.16 | 53.67 ± 2.00 | 177.33 ± 3.0 | 74.71 ± 5.25 | 9.51 ± 1.50 | 30.59 ± 3.87 | 50.40 ± 4.68 | 157.85 ± 4.46 |
| COS 50 | 87.67 ± 2.08 | 10.33 ± 1.52 | 32.00 ± 2.64 | 57.33 ± 3.05 | 189.33 ± 2.5 | 85.91 ± 2.04 | 10.19 ± 1.44 | 31.25 ± 2.46 | 57.86 ± 1.60 | 183.67 ± 4.33 |
| OGA 50 | 86.67 ± 7.63 | 10.33 ± 0.57 | 33.33 ± 2.88 | 57.67 ± 3.78 | 188.00 ± 8.0 | 84.67 ± 7.83 | 10.09 ± 0.59 | 32.55 ± 2.73 | 56.88 ± 3.39 | 178.57 ± 6.88 |
| AOS 50 | 91.33 ± 1.15 | 10.67 ± 0.57 | 34.33 ± 3.51 | 59.00 ± 2.00 | 191.00 ± 4.6 | 90.41 ± 0.58 | 10.56 ± 0.66 | 33.89 ± 3.62 | 59.98 ± 1.74 | 187.15 ± 2.65 |
| Level of significant | * | NS | NS | * | * | * | NS | NS | NS | ** |
| LSD (0.05%) | 13.57 | – | – | 8.69 | 16.15 | 10.34 | – | – | – | 10.88 |

**Notes:**
[X]All treatments were applied at 15, 30, 45 and 60 days after transplanting @ 50 mg.L$^{-1}$ except control (sterile water); COS, chitosan oligosaccharide; OGA, oligo galacturonic acid; AOS, alginate oligo saccharide; values are as the means ± SD in two independent measurements; LSD, least significance difference value at 0.05%; NS, not significant; asterisks indicate significant differences (*$P < 0.05$, **$P < 0.01$).
[Y]cm, centimetre.

Visually abundant early flowering and pods were observed in the OSs-treated plants compared to the control (Fig. 2). Only pods filled with seeds were considered for counting. A significantly increased number of pods was spotted in the OSs-treated plants compared to the control group in both seasons (Table 3). Additionally, a trend of total pods was obtained from AOS>COS>OGA among the OSs treated in both seasons (Fig. 2). Meanwhile, the maximum number of total pods detected in AOS-treated plants was 191.00 and 187.15 in early and mid-season, respectively.

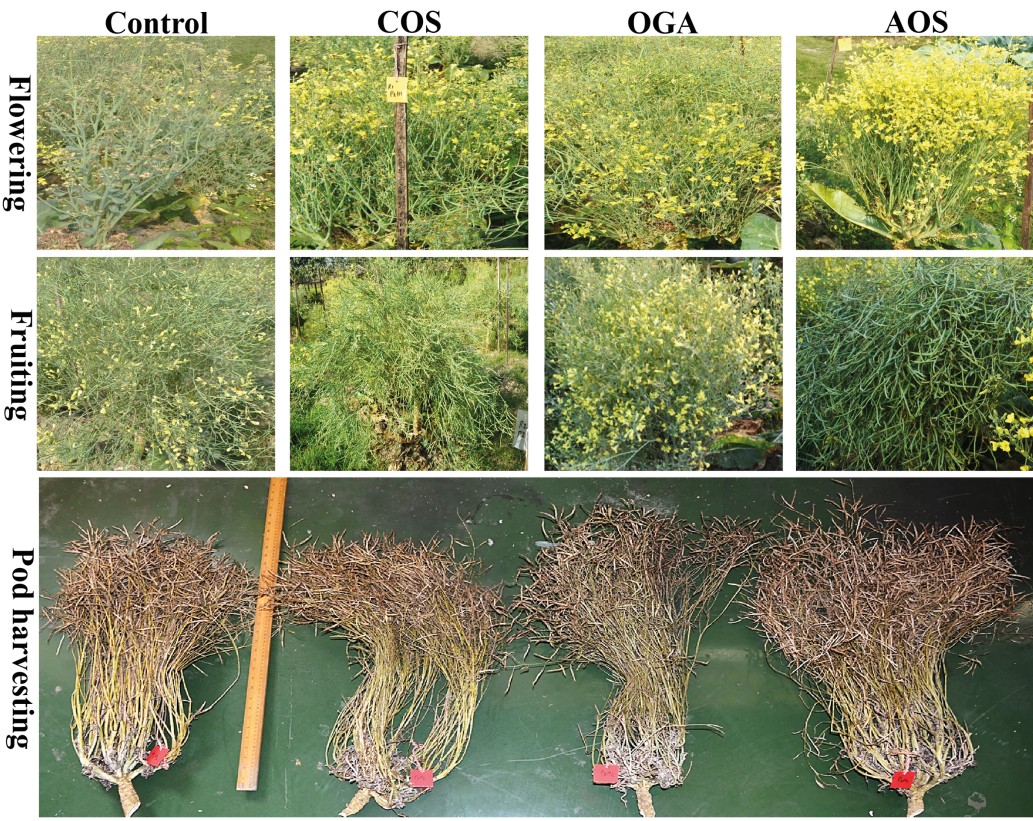

**Figure 2 Flowering, fruiting and total pod along with stalk length and branches of BU cauliflower 1 as influenced by different oligosaccharides.** All treatments were applied at 15, 30, 45 and 60 days after transplanting @ 50 mg.L$^{-1}$ except control (sterile water); COS, chitosan oligosaccharide; OGA, oligo galacturonic acid; AOS, alginate oligosaccharide.

## Seed yield induced by oligosaccharides

The early-season experiment exhibited statistically higher values in OSs-treated plants compared to the control, while no significant difference was recorded in the mid-season experiment (Table 4). The best pod length was noted from OGA (6.25 cm) and AOS (6.20 cm) treated plants in the early season and mid-season, respectively. AOS exhibited the highest pod diameter (0.45 and 0.41 mm) in both seasons. However, after being treated by three OSs, pod length and diameter did not exhibit statistically significant differences.

The number of seeds per pod was significant in the early season and non-significant in the mid-season after being treated by three different OSs compared to the control. Although a comparatively higher number of seeds was detected in pods, the variation among the OSs treated was uniform. The data showed that COS produced the maximum number of seeds (15.07) in the early and (14.82) mid-season, followed by AOS>OGA (Table 4).

Total seeds harvested from the individual plants were weighed before drying and expressed as seed per plant (gram weight basis, g$^{wb}$). Seed weight varied significantly at both seasons among the treatments (17.8–64.5%), and OSs-treated plants produced more (Table 4). AOS-treated plants yielded the maximum seeds (54.23 and 51.54 g$^{wb}$) in early

**Table 4  Seed yield attributes of BU cauliflower 1 under different oligosaccharides treatments.**

| Treatment[X] | Early Season (15 September 2021)[Y] | | | | Mid-Season (15 October 2021) | | | |
|---|---|---|---|---|---|---|---|---|
| | Pod length (cm) | Pod diameter (mm) | No. of seed/pod | Seed/plant (g) | Pod length (cm) | Pod diameter (mm) | No. of seed/pod | Seed/plant (g) |
| Control | 4.81 ± 0.07 | 0.26 ± 0.01 | 13.77 ± 0.85 | 34.57 ± 0.60 | 5.14 ± 0.60 | 0.26 ± 0.01 | 13.09 ± 1.98 | 31.33 ± 0.54 |
| COS 50 | 5.88 ± 0.58 | 0.35 ± 0.02 | 15.07 ± 1.90 | 40.73 ± 6.12 | 5.80 ± 0.57 | 0.31 ± 0.06 | 14.82 ± 3.09 | 39.44 ± 5.18 |
| OGA 50 | 6.25 ± 0.30 | 0.41 ± 0.12 | 14.57 ± 2.25 | 43.20 ± 3.90 | 6.08 ± 0.32 | 0.37 ± 0.15 | 14.12 ± 2.08 | 41.60 ± 3.60 |
| AOS 50 | 6.20 ± 0.54 | 0.45 ± 0.05 | 14.70 ± 0.90 | 54.23 ± 10.54 | 6.20 ± 0.57 | 0.41 ± 0.13 | 14.58 ± 2.14 | 51.54 ± 11.91 |
| Level of significant | * | * | * | * | NS | NS | NS | * |
| LSD (0.05%) | 1.23 | 0.16 | 4.62 | 15.27 | – | – | – | 12.56 |

Notes:
[X]All treatments were applied at 15, 30, 45 and 60 days after transplanting @ 50 mg.L$^{-1}$ except control (sterile water); COS, chitosan oligosaccharide; OGA, oligo galacturonic acid; AOS, alginate oligo saccharide; values are as the means ± SD in two independent measurements; LSD, least significance difference value at 0.05%; NS, not significant, asterisks indicate significant differences (*$P < 0.05$).
[Y]cm, centimetre; mm, millimetre; g, gram.

(56.8%) and mid-season (64.5%). The seed yield accelerated in early (56.8%) and mid-season (64.5%) compared to the control. The trend of seed weight per plant was AOS>OGA>COS and was statistically similar among them, but only AOS significantly differed from the control in both seasons.

## Physiological traits induced by oligosaccharides

The SPAD value directly indicates chlorophyll content and photosynthetic activity (*Zhang et al., 2022*). SPAD value was detected in the leaves before and after the final OSs application. A significant increase of 2.5–3.3% in SPAD values was observed in OS-treated plants compared to the control (0.6%) in both seasons. The order of SPAD values was statistically similar, with COS>AOS>OGA (3.3, 2.75, and 2.5%) significantly differing from the control (Fig. 3A and Table S3). The chlorophyll content in leaves was also quantified to validate SPAD values. In line with the SPAD value trend, a similar pattern was observed in leaf chlorophyll content following OSs application. The highest chlorophyll content was estimated in COS-treated plant leaves (2.5 mg.g$^{-1}$ FW) in the early season and AOS-treated plants (2.54 mg.g$^{-1}$ FW) in the mid-season, both significantly diverging from the control but not from each other. However, the value obtained from OGA (2.0–2.1 mg.g$^{-1}$ FW) was not significant from the control (1.6–1.7 mg.g$^{-1}$ FW) (Fig. 3B and Table S3).

## Plant defense induced by oligosaccharides

Brassicaceae vegetables are frequently infected by black rot, attributed to the necrotrophic bacterial pathogen *Xanthomonas campestris* pv. *Campestris*, resulting in severe consequences related to quality and yield losses (*Liu et al., 2022*). Applying AOS and COS significantly enhanced disease resistance to black rot from phenotypic and disease index perspectives compared to the control group. However, although OGA appeared to improve disease resistance, the effect was insignificant compared to the control in both seasons (Fig. 4 and Table S3). Meanwhile, AOS-treated plants possessed the highest

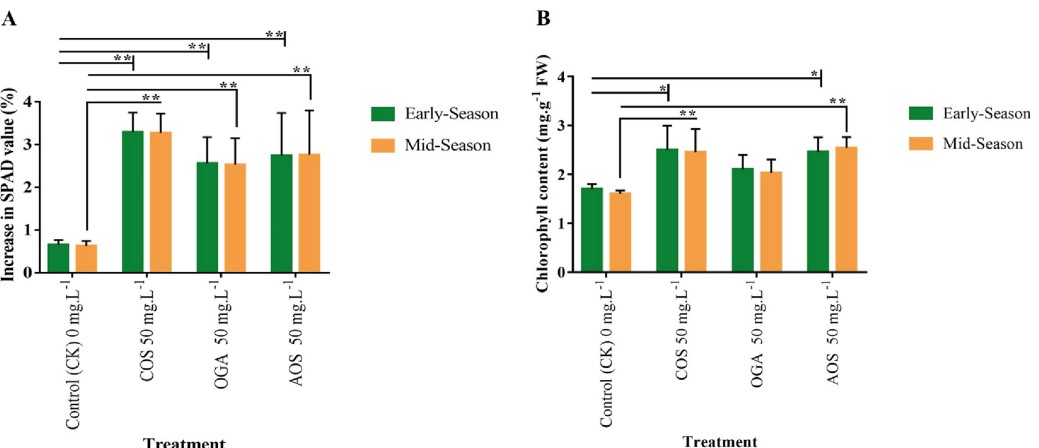

**Figure 3 SPAD value and chlorophyll content in leaves of BU cauliflower 1 as influenced by different oligosaccharides.** (A) SPAD value. (B) Chlorophyll content. All treatments were applied at 15, 30, 45 and 60 days after transplanting @ 50 mg.$L^{-1}$ except control (sterile water); COS, chitosan oligosaccharid; OGA, oligo galacturonic acid; AOS, alginate oligosaccharide). SPAD was recorded with the help of SPAD-502 Plus chlorophyll meter (Konica-Minolta, Japan) 5 days before and after the 4th spraying from the same leaves. Subsequently chlorophyll content was detected in the laboratory. Asterisks indicate significant differences ($^*P < 0.05$, $^{**}P < 0.01$).

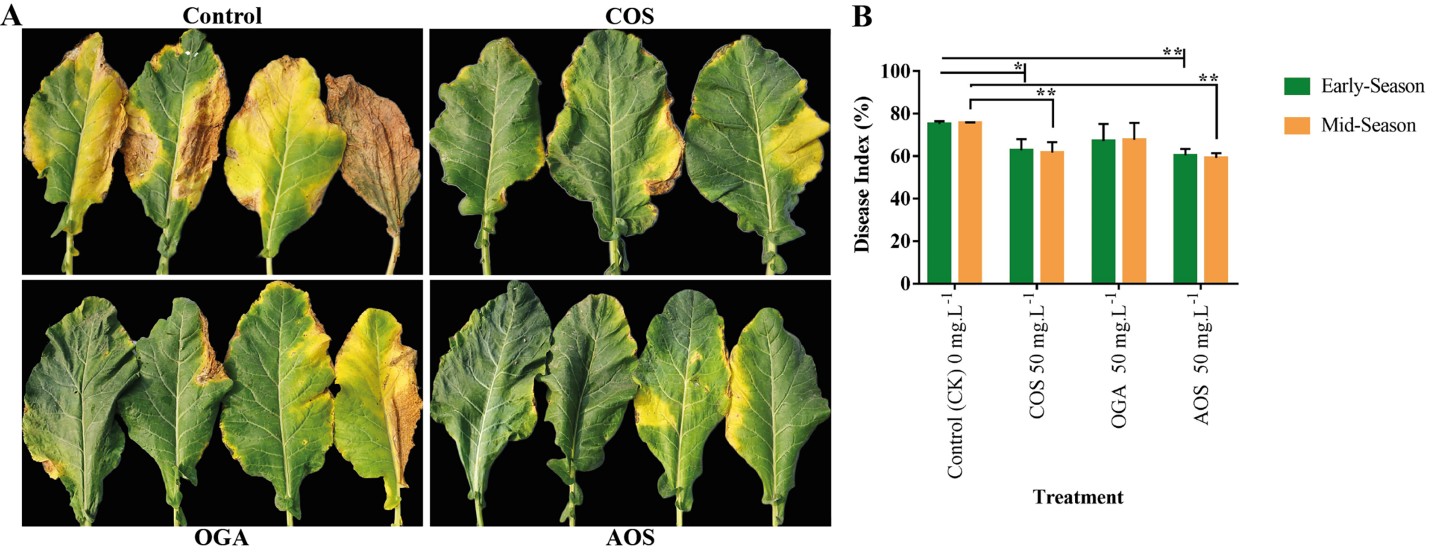

**Figure 4 Oligosaccharides enhanced disease resistance to black rot in BU cauliflower 1.** (A) Black spot severity. (B) Disease index. All treatments were applied at 15, 30, 45 and 60 days after transplanting @ 50 mg.$L^{-1}$ except control (sterile water), COS = chitosan oligosaccharide, OGA = oligo galacturonic acid, AOS = alginate oligo saccharide), asterisks indicate significant differences ($^*P < 0.05$, $^{**}P < 0.01$).

immunity in the early and mid-season (60.41 and 59.18%), followed by COS (62.78 and 61.71%, respectively).

Free radicals, along with signaling molecules reactive oxygen species (ROS), represent the earliest hallmark signaling molecules correlated with plant defense resistance (*Lee et al., 2020*). In contrast to the control, AOS and OGA-treated plant leaves significantly increased higher $H_2O_2$ quantities. The boosting pattern was AOS>OGA, with AOS

**Table 5 Secondary metabolites of BU cauliflower 1under different oligosaccharides treatments.**

| Treatment[X] | Early season (15 September 2021)[Y] | | | | | | Mid-season (15 October 2021) | | | | | |
|---|---|---|---|---|---|---|---|---|---|---|---|---|
| | $H_2O_2$ (mg.g$^{-1}$ FW) | MDA (nmol. g$^{-1}$ FW) | Total phenol (mg.100g$^{-1}$ FW) | Total flavonoid (QE.100 g$^{-1}$ FW) | Ascorbic acid (mg.100 g$^{-1}$) | IC$_{50}$ (µg.mL$^{-1}$) | $H_2O_2$ (mg.g$^{-1}$ FW) | MDA (nmol. g$^{-1}$ FW) | Total phenol (mg.100 g$^{-1}$ FW) | Total flavonoid (QE.100 g$^{-1}$ FW) | Ascorbic acid (mg.100 g$^{-1}$) | IC$_{50}$ (µg.mL$^{-1}$) |
| Control | 0.86 ± 0.1 | 4.40 ± 0.1 | 14.34 ± 1.3 | 8.34 ± 0.2 | 13.2 ± 0.6 | 540.32 ± 30 | 0.82 ± 0.1 | 4.17 ± 0.1 | 13.87 ± 1.4 | 7.92 ± 0.2 | 12.75 ± 0.5 | 500.90 ± 36 |
| COS 50 | 0.92 ± 0.0 | 3.44 ± 0.3 | 24.79 ± 1.1 | 8.75 ± 0.3 | 16.72 ± 0.7 | 340.91 ± 20 | 0.95 ± 0.1 | 3.34 ± 0.3 | 24.30 ± 1.3 | 8.57 ± 0.2 | 16.44 ± 0.6 | 324.00 ± 22 |
| OGA 50 | 1.78 ± 0.3 | 4.02 ± 0.1 | 18.09 ± 2.1 | 8.93 ± 0.1 | 14.08 ± 0.3 | 432.33 ± 12 | 1.73 ± 0.3 | 3.93 ± 0.1 | 18.19 ± 1.8 | 8.75 ± 0.1 | 14.03 ± 0.3 | 415.00 ± 10 |
| Level of significant | *** | ** | ** | * | *** | *** | *** | ** | *** | ** | *** | *** |
| LSD (0.05%) | 0.57 | 0.59 | 4.63 | 0.73 | 1.39 | 52.69 | 0.42 | 0.39 | 3.07 | 0.43 | 0.86 | 45.38 |

Notes:
[X] All treatments were applied at 15, 30, 45 and 60 days after transplanting @ 50 mg.L$^{-1}$ except control (sterile water); COS, chitosan oligosaccharide; OGA, oligo galacturonic acid; AOS, alginate oligo saccharide; values are as the means ± SD in two independent measurements; LSD. least significance difference value at 0.05%, asterisks indicate significant differences (*$P < 0.05$, **$P < 0.01$, ***$P < 0.001$).
[Y] mg, milligram; µg, microgram; g$^{-1}$, per gram; nmol, nanomole; QE, quercetin equivalents; FW, fresh weight basis; $H_2O_2$, hydrogen per oxide; MDA, malondialdehyde; IC$_{50}$, 50 percent inhibition concentration.

generating 2.22 and 2.15 mg.g$^{-1}$ FW, while OGA generated 1.78 and 1.73 mg.g$^{-1}$ FW of $H_2O_2$ during the early and mid-season (Table 5). Interestingly, COS also released a comparatively higher amount of $H_2O_2$ at early and mid-season (0.92 and 0.95 mg.g$^{-1}$ FW) but it was not significantly higher than the control.

A high level of malondialdehyde (MDA) indicates oxidative damage to cellular components, a marker of lipid peroxidation. MDA content can, therefore, be used as an indicator of oxidative stress damage (*Chaoui et al., 1997*). A significantly higher amount of MDA was found in control plants' leaves than in OSs treated plants. The dominance pattern was Control>OGA>COS>AOS in both seasons. Higher accumulation of MDA in control plants (4.40 and 4.17 nmol.g$^{-1}$ FM of MDA) was detected in the early and mid-season, respectively (Table 5). AOS-treated plants accumulated the least amount of MDA in the early and mid-season (3.39 and 3.32 nmol.g$^{-1}$ FM), which was identical to COS treatments. This indicated that OSs applied to cauliflower could cause less membrane lipid damage after being infected with black rot, whereas AOS and COS could perform better.

Phenolics are important compounds for plant development, structural integrity, skeleton support, and secretion of defense molecules against biotic and abiotic stresses (*Bhattacharya, Sood & Citovsky, 2010*). Our study estimated a significantly higher amount of total phenol under OSs treatments than the control. The total phenol content showed a trend of COS>AOS>OGA in both seasons among OSs treatments. COS generated total phenol (24.79 and 24.30 mg 100g$^{-1}$ DW) in early and mid-season experiments, where AOS delivered a statistically similar quantity of phenols to COS (Table 5). On the other hand, OGA also released a higher quantity of phenols to the control, but it was significant only at mid-season and not at early season. These results indicated that OSs could generate total phenols under stress.

Flavonoids are diverse secondary metabolites that participate in many functions like plant development, pigmentation, signaling, and protection against various pathogens (*Mathesius, 2018*). A significantly higher level of flavonoids was also identified when OSs

were applied to plants at mid-season, while it was only significant in AOS treatment in the early season. AOS assembled the highest levels (9.31 and 9.24 QE.100g$^{-1}$ DW) in the respective two seasons of experiments, followed by OGA>COS. This variation of flavonoids at mid-season was more prominent with the control than early season, indicating OSs ability to generate flavonoids under stress, with AOS being the most effective (Table 5).

Ascorbic acid is also considered as a non-enzymatic antioxidant that neutralizes ROS overproduction and contributes to several physiological functions in plants (*Akram, Shafiq & Ashraf, 2017*). The lowest amount of ascorbic acid was detected in control plants, while OSs treated plants exhibited significantly higher levels in both experiments (except OGA in the early season). The accumulation trend of ascorbic acid was AOS>COS>OGA in OSs treated plants in both seasons. AOS stimulated 16.72 and 16.71 mg.100g$^{-1}$ FW of ascorbic acid in the early and mid-season (Table 5). COS performance was statistically similar to AOS, while OGA was distinctly lower than AOS and COS. These results indicated that OSs successfully generated non-enzymatic antioxidants under black rot and fluctuating growing seasons, with AOS and COS being comparatively better.

A half maximal concentration (IC$_{50}$) value is the concentration of the respective sample that scavenges 50% of the DPPH free radical. It determines antioxidant effectiveness against specific biological and biochemical functions. The lower concentration indicates the higher effectiveness of antioxidant activity (AOA) to scavenge free radicals and *vice-versa* (*Dhanani et al., 2017*). Significantly lower concentrations (IC$_{50}$) were detected in OSs treated plants compared to the control. Antioxidant activity was AOS < COS < OGA, with significant differences. The lowest concentration was found in AOS-treated plants (115.49 and 119.31 µg.mL$^{-1}$) in early and mid-season experimented plants, while the control was 540.32 and 500.90 µg.mL$^{-1}$, respectively (Table 5). These results indicated that OSs had higher adeptness at scavenging free radicals with efficient antioxidant activity in plant immune-related biological functions.

## Correlation matrix and cluster analysis of studied parameters

The correlation matrix between the studied parameters showed moderate to strong positive and negative relationships with one or more (Fig. 5A). As observed, MDA was strongly positively correlated with IC$_{50}$ (AOA), while IC$_{50}$ was strongly linked with DI than MDA. Nonetheless, MDA, IC$_{50}$, and Disease index (DI) did not appear to have any positive relationship with all other studied variables. The MDA was more negative when it came to ascorbic acid>total phenol>leaf length>SPAD>curd yield>tertiary branch>chlorophyll>seed number per plant. The IC$_{50}$ was convincingly negative for secondary metabolites such as ascorbic acid, flavonoids, phenols, H$_2$O$_2$, and even seed yield. The DI had a strong negative relation with seed yield and yield attributes but less correlated with morphological and yield attributes. The DI also had a high negative relation to ascorbic acid, phenol, and flavonoids content.

Plant height was positively related to leaf characters (number, length, and breadth), marketable yield, curd diameter, and SPAD. Curd yield is deeply correlated with curd length, SPAD and leaf breadth, phenols, ascorbic acid, and flavonoids. However, stalk

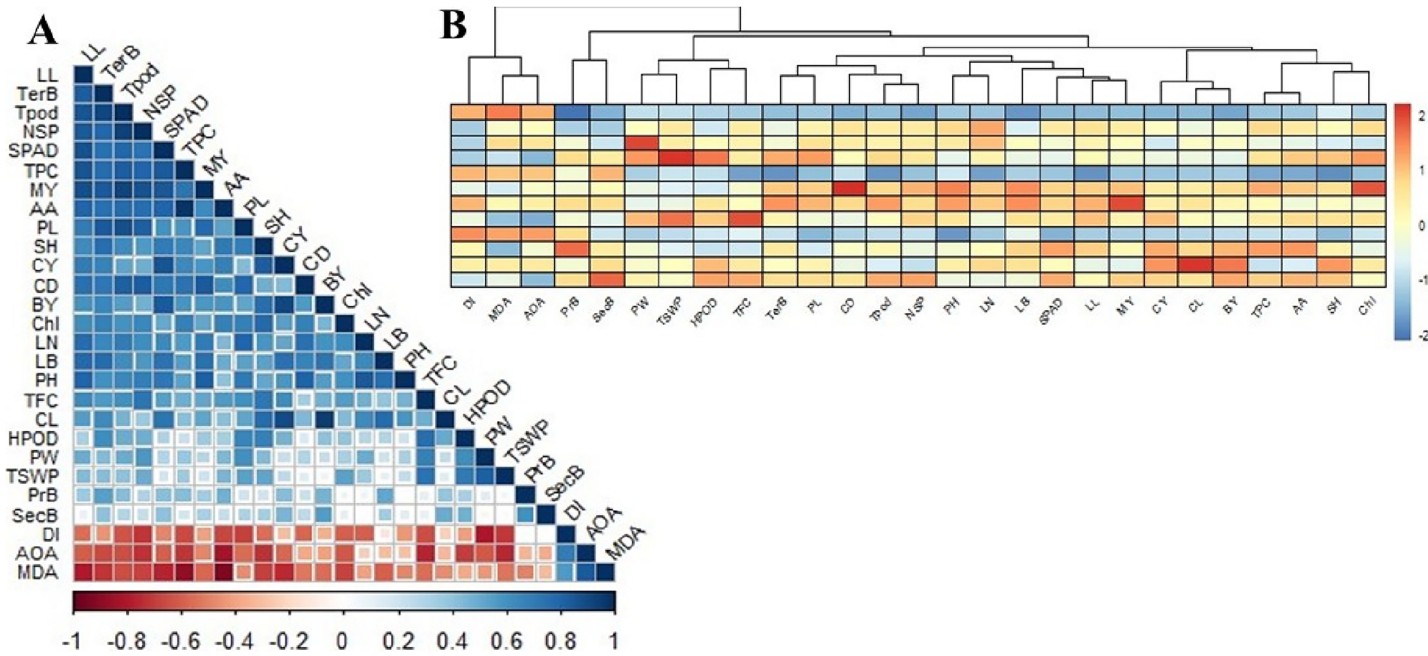

**Figure 5 Correlation matrix and heat map among the studied morphophysiological and yield attributes, secondary metabolites and disease index of BU cauliflower 1 under different oligosaccharides.** (A) Correlation matrix. (B) Heat map. LL, leaf length; TerB, tertiary branch; Tpod, Total pod; NSP, number of seed per po; SPAD, percent SPAD value increased; TPC, total phenols content; MY, marketable yield; AA, ascorbic acid; PL, pod length; SH, stalk height; CY, curd yield; CD, curd diameter; BY, biological yield; Chl, chlorophyll, LN, leaf number; LB, leaf breadt;, PH, plant height; TFC, total flavonoids content; CL, curd length; HPOD, hydrogen per oxide; PW, pod diameter; TSWP, seed per plant; PrB, primary branch; SecB, secondary branch; DI, disease index; AOA, antioxidant activity ($IC_{50}$ value); MDA, malondialdehyde.

height was dependent on curd yield (CL>CD) and the tertiary branch was linked to stalk height. Meanwhile, the total pod was highly dependent on the tertiary branch, leaf length, and curd diameter. Seed yield was more related to pod diameter than pod length and was highly associated with flavonoids and $H_2O_2$. There was also a profound positive correlation between physiochemical and secondary metabolites. SPAD and chlorophyll were positively correlated with phenols and ascorbic acid content, while phenols and ascorbic acid correlation was the strongest. Flavonoids were intensely correlated with $H_2O_2$.

Based on the correlation, there were two main clusters with distinct deviations from each other (Fig. 5B). Cluster-I includes MDA, $IC_{50}$, and DI; cluster-II is grouped with the rest of the studied parameters. Cluster II was further sub-grouped into two major groups, where curd yield and yield components (especially curd length) were grouped with morphological attributes, SPAD, chlorophyll, phenol, and ascorbic acid. Meanwhile, seed yield and yield attributes were grouped with curd diameter, $H_2O_2$, and flavonoids. From these cluster analyses, it could be summarized that curd yield was reliant on curd length. Seed yield was correlated with curd diameter accompanied by distinguished secondary metabolites, while MDA, $IC_{50}$, and DI were negatively correlated with the rest of all 24 studied variables (Fig. 5B).
## Principal component analysis of OSs

Principal component analysis (PCA) was employed to depict the relationship and impact of different OSs on the morphology, curd, seed yield, secondary metabolites, and disease index with precise evaluation. PCA allowed us to figure out how certain variables were interconnected to categorize the OSs. For this, two significant PCs (Dim1 and Dim2) with an overall explained variance of 69.4% using the 27 variables were generated (Figs. 6A and 6B). From the variables-PCA plot, it has been seen that the first PC1 (Dim1) explained 57.9% of total variables with positive loading of 24 variables and three variables (MDA, $IC_{50}$, and DI) in the negative loading site. Meanwhile, 11.5% of the total variability was predicted by PC2 (Dim2) with positive loading of 17 variables, and the rest of the 10 variables had a negative loading of PC2 (Dim2). Based on both loading sites, leaf breadth, SPAD/chlorophyll, curd diameter and length, curd yield, tertiary branch, total pod, seed per pod, seed yield, total phenol, ascorbic acid, flavonoids, MDA, $IC_{50}$, and DI could be prioritized for similar studies in the future.

The PCA-Biplot consists of PC1 vs. PC2 (Dim1 vs. Dim2) drawn as vectors, which showed that three oligosaccharides (COS, OGA, AOS) and control treatments occupied different regions of the plot with well-defined patterns (Fig. 6B). Biplot simultaneously represents both the imposed treatments and the variables with a specific direction along with the specific PC axis. Variable arrows indicate the path along which the corresponding variable's contribution increases the most, and their lengths represent the rate of change. The clustering of different OS treatments, including control (water), shown in the ellipse, indicated that cauliflower plants were more prone to disease with the maximum MDA and $IC_{50}$ value when not treated with OSs (control). These plants fell in the opposite direction to the AOS, COS, and OGA concerning the PC1. By analyzing PC2, AOS-treated plants were located on the opposite side of MDA, $IC_{50}$ value, and DI with a high contribution of seed yield and its related attributes, including ascorbic acid, flavonoids, and moderate contribution of $H_2O_2$ content. Likewise, COS>OGA treated plants were grouped into positive loading of PC1 and PC2 with the highest contribution to morphological and related field-related attributes such as lea breadth, curd length, and diameter, curd yield, marketable yield, SPAD, total phenol along with the moderate contribution of chlorophyll. These results indicated that cauliflower seed yield was augmented by applying AOS with less MDA, $IC_{50}$ value, and disease. Secondary metabolites such as ascorbic acid, flavonoids, and $H_2O_2$ contributed the most in this regard. Meanwhile, COS and OGA enriched morphology and curd yield with less MDA, $IC_{50}$ value, and disease mainly through the higher contribution of SPAD, chlorophyll, and total phenol.

## DISCUSSION

The pursuit of an optimal temperature range (10–15 °C) is imperative for the growth, curd formation, and seed yield of cauliflower (*Brassica oleracea var. botrytis* L.) (*Din et al., 2007*). Fluctuations in temperature, whether elevated or decreased, hamper curd formation and seed yield (*Sharma et al., 2020*). In Bangladesh, the persistent challenge of low curd and seed yield is attributed to temperature constraints (*Prodhan et al., 2022*) and the notorious black rot caused by *Xanthomonas campestris* pv. *Campestris in* cauliflower

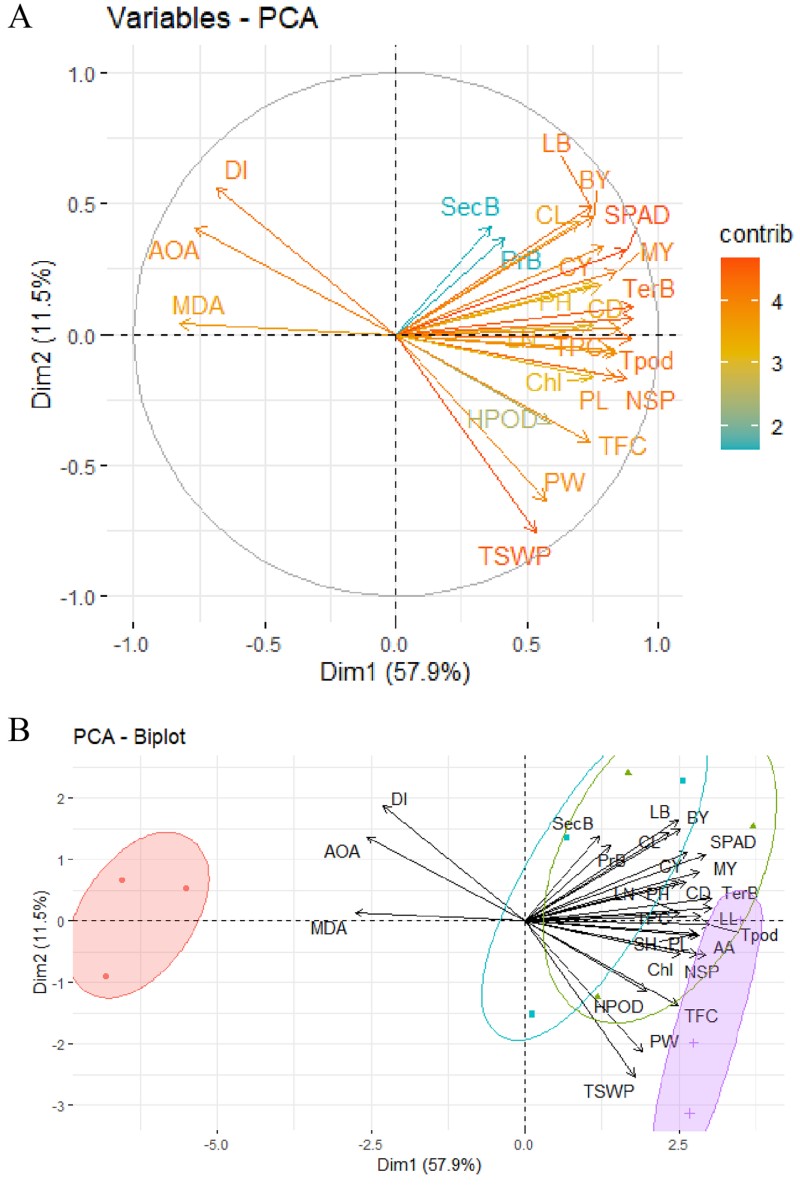

**Figure 6 PCA of different major morpho-physiological and yield attributes, secondary metabolites and disease index in BU cauliflower 1 under different oligosaccharides.** (A) Variables-PCA. (B) PCA-biplot. P1, control (water); P2, chitosan oligosaccharide (COS); P3, oligo galacturonic acid (OGA); P4, alginate oligosaccharide (AOS); LL, leaf length; TerB, tertiary branch; Tpod, Total pod; NSP, number of seed per pod; SPAD, percent SPAD value increased; TPC, total phenols content; MY, marketable yield; AA, ascorbic acid; PL, pod length; SH, stalk height; CY, curd yield; CD, curd diameter; BY, biological yield; Chl, chlorophyll; LN, leaf number; LB, leaf breadth; PH, plant height; TFC, total flavonoids content; CL, curd length; HPOD, hydrogen per oxide; PW, pod diameter; TSWP, seed per plant; PrB, primary branch; SecB, secondary branch; DI, disease index; AOA, antioxidant activity (IC$_{50}$ value); MDA, malondialdehyde.

(*Li et al., 2019*). The average temperature ranges from 25–30 °C, with rainfall and dew facilitating its penetration and subsequent infection (*Vicente & Holub, 2013*). Although there was a lower seed yield compared to temperate countries, a promising variety of BU

cauliflower one was employed to detect the effects of different oligosaccharides (OSs) applications. Two experiments, conducted in early and mid-season trials, aimed to elucidate the impact of OSs on temperature variations during the growing season. We unveiled significant enrichments in biomass, curd and seed yield, physiochemical and secondary metabolites, coupled with a reduction in MDA, $IC_{50}$, and disease index when pre-treated with OSs compared to control. Notably, the early season was superior to its mid-season counterpart. All studied parameters suffered more in mid-season than in early season, which might be due to unpredictable environmental fluctuations during the two studied seasons.

OGA emerged as influential in enhancing plant height, leaf number, and leaf breadth in the early season among the three OSs but did not sustain its effectiveness in the mid-season, where COS took over. COS demonstrated proficiency in promoting leaf length along with biological, marketable, and curd yield. Its effectiveness lies in the elevated SPAD value, chlorophyll, phenols, and ascorbic acid content under fluctuating temperature conditions. Although OGA claimed superiority in curd (length and diameter) in both seasons, its size was statistically identical to COS-treated plants. The superior vegetative growth, yield, and physical-nutrition curd quality from OGA and COS could be attributed to a higher elevation of chlorophyll and fostering greater photosynthetic activity leading to the synthesis of various metabolites (*Sheikha & Al-Malki, 2011*; *Rahman et al., 2018*; *Marzouk, Abd-Alrahman & El-Sawy, 2022*). Chitosan, due to its amino components and degradation in the soil, facilitated the production of excellent tubers in kohlrabi along with maximum yields and leaves containing a rich nutrient composition (*El-Bassiony et al., 2014*). Chitosan also played a crucial role in generating endogenous phytohormones such as gibberellic acid and auxin in navel orange (*Ahmed et al., 2016*), potentially contributing to better growth and development (*Uthairatanakij, Teixeira da Silva & Obsuwan, 2007*). Several experiments corroborate the greater function of COS concerning biomass, physiochemical, and secondary metabolites in different plants, providing direct support to the present study (*Ahmed et al., 2020*; *Jia et al., 2020*).

AOS improved phenological characteristics, seed yield (56.8–64.5%), and most secondary metabolites (except MDA and $IC_{50}$). AOS exhibited the lowest MDA and $IC_{50}$ value with alleviated disease symptoms and enhanced resistance to black rot disease compared with the control, surpassing even COS and OGA in both seasons. Higher levels of $H_2O_2$, flavonoids, and ascorbic acid were detected in plants treated with AOS (50 mg.$L^{-1}$) in both seasons. Likewise, higher total phenols and flavonoids were detected in strawberries (AOS 100 mg.$L^{-1}$) (*Bose et al., 2019*), while *Eucomis autumnalis* treatment with AOS exhibited higher L-ascorbic acid (*Salachna et al., 2018*). Plants adapted enzyme-dependent or independent mitigation and detoxification systems against ROS during stress conditions. AOS can activate the antioxidant defense system against ROS injury, as evidenced by increased catalase (CAT), superoxide dismutase (SOD), and peroxidase (POD) in rice seedlings sprayed with AOS (DP 13.9, at 3,000 µg.$mL^{-1}$) (*Qiao & Ouyang, 2013*). Similar tales of the resilience of MDA and proline were reported in wheat in combating Cadmium drought (1,000 mg.$L^{-1}$ of alginate-derived OS) (*Ma, Li & Bu, 2010*). *Hernández-Herrera et al. (2014)* affirmed that guaiacol peroxidase and polyphenol

oxidase were enhanced (0.1 mg.mL$^{-1}$), both related to protecting tomato from *Alternaria solani*. AOS at 25 mg.L$^{-1}$ accelerated ROS (H$_2$O$_2$) and NO early signaling molecules production in Arabidopsis through an increase in salicylic acid (SA) and defense gene *PR1*, reducing the disease index caused by Pst DC3000 (*Zhang et al., 2019*). Therefore, AOS's impact culminated in a remarkable enhancement in seed yield by 56.8% and 64.5% in early and mid-season, respectively, through enhancing plant immunity-related metabolites with reduced MDA, IC$_{50}$, and disease index. Similarly, *Mannan et al. (2023)* reported 54.87% and 23.97% soybean yields under drought and wet conditions by applying 10% seaweed extract.

Additionally, a correlation matrix and Principal Component Analysis (PCA) functioned as navigational tools for uncovering the most promising factors exposed to OS application for superior cauliflower curd and seed production. The correlation matrix elucidated positive alliances among MDA, IC$_{50}$, and disease index but revealed negative associations with other studied parameters, notably with ascorbic acid, total flavonoids, and total phenols. Curd yield formed positive connections with these three and extended its affections to curd length, SPAD, and chlorophyll. Seed yield correlated with curd diameter, flavonoids, and H$_2$O$_2$. Strong positive correlations were observed between total phenol - ascorbic acid and flavonoids - H$_2$O$_2$. Plant biomass correlated with SPAD and chlorophyll, and these two were related to phenols and ascorbic acid. Principal component analysis (PCA) depicted and clustered flavonoids, ascorbic acid, and H$_2$O$_2$ as major contributors to seed yield, which was improved by AOS (50 mg.L$^{-1}$). On the other side, plant biomass, SPAD/chlorophyll, and total phenols were recognized as the major contributors to curd yield, which was enhanced by COS>OGA (50 mg.L$^{-1}$). MDA, IC$_{50}$ value, and DI were mentioned as opposite major contributors to all previously mentioned parameters, with the control group lingering close to these sites and the AOS group standing at the pinnacle, a beacon of safety and efficiency.

## CONCLUSION

Significantly better biomass, curd, and seed yield, physiochemical and secondary indicators emerged from different OS treatments, eclipsing the control. Comparatively advantageous results were discerned in the early season than in the mid-season due climate fluctuation during two growing seasons, albeit with a consistent achievement trend. The control group lagged behind, suffering more in all studied parameters in the mid-season compared to the early season. These findings provide evidence of the efficacy of OSs on biomass, curd, and seed yield of cauliflower under stress conditions. COS and OGA (50 mg.L$^{-1}$) enhanced plant biomass and curd yield (5–14.55%) based on improvement through SPAD, chlorophyll, and total phenols content. Likewise, AOS (50 mg.L$^{-1}$) boosted seed yield by 56.8–64.5% and enhanced plant immunity with superior secondary metabolites, reducing MDA, IC$_{50}$, and disease index. Ascorbic acid–phenols, flavonoids-H$_2$O$_2$, and SPAD/chlorophyll-ascorbic/phenols were positively correlated and played a crucial role in plant immunity, curd, and seed production of cauliflower. Therefore, this study suggests for considering COS, OGA for biomass and curd yield, and AOS for seed production as a safeguard against biotic and abiotic stress in the future. Despite conducting multiple

locations and trials in multiple seasons, this study did not examine abiotic stress against specific stressors. Further investigation is needed to investigate the molecular mechanisms behind the cascade of OS applications regulating cauliflower curd and seed production.

### Funding

This work was supported by the ANSO Collaborative Research Program (ANSO-CR-KP-2020-14), and the National Natural Science Foundation of China (32000203 and 32000905). Md. Mijanur Rahman Rajib was supported by the DICP, CAS PhD program. The funders had no role in study design, data collection and analysis, decision to publish, or preparation of the manuscript.

### Grant Disclosures

The following grant information was disclosed by the authors:
ANSO Collaborative Research Program: NSO-CR-KP-2020-14.
National Natural Science Foundation: 32000203 and 32000905.
DICP, CAS PhD program.

### Competing Interests

Heng Yin is an Academic Editor for PeerJ.

### Author Contributions

- Md. Mijanur Rahman Rajib conceived and designed the experiments, performed the experiments, analyzed the data, prepared figures and/or tables, authored or reviewed drafts of the article, and approved the final draft.
- Hasina Sultana conceived and designed the experiments, performed the experiments, analyzed the data, authored or reviewed drafts of the article, and approved the final draft.
- Jin Gao analyzed the data, prepared figures and/or tables, authored or reviewed drafts of the article, and approved the final draft.
- Wenxia Wang conceived and designed the experiments, authored or reviewed drafts of the article, contributed reagent, and approved the final draft.
- Heng Yin conceived and designed the experiments, authored or reviewed drafts of the article, and approved the final draft.

### Data Availability

The raw measurements are available in the Supplemental Files.

### Supplemental Information

Supplemental information for this article can be found online at http://dx.doi.org/10.7717/peerj.17150#supplemental-information.

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
