# Peer review of "Curd, seed yield and disease resistance of cauliflower are enhanced by oligosaccharides"

_PeerJ, doi:10.7717/peerj.17150_

## Round 0.1 · original submission · Major Revisions

Please revise the article considering the reviewers' comments, especially reviewer 1.

Please note that the article is on the margin of rejection.

**Language Note:** The review process has identified that the English language must be improved. PeerJ can provide language editing services - please contact us at copyediting@peerj.com for pricing (be sure to provide your manuscript number and title). Alternatively, you should make your own arrangements to improve the language quality and provide details in your response letter. – PeerJ Staff

·

Basic reporting

The experiment is well designed and executed. Non-hazardous oligosaccharides from natural sources have potential to use in managing crop health particularly at harvest stage when chemical measures are prohibited. The study highlights their use in cauliflower, it seems interesting.

Experimental design

In Materials & Methods section, I suggest to add the reference for Disease index calculation. It seems that authors followed some existing scale and formula.
It is suggested to write the correct reference for the Folin-Ciocalteu method, it is fron from Mohammed, Edna & Siraj, 2020. Also, explain DPPH and similar abbreviation in first place.

The reported disease is not appears to be Alternaria leaf but it is black rot (as visible in both plant & leaf based pics)
So authors should change their presentation/article accordingly.

Validity of the findings

The findings need to be presented in the context of black rot.
Antioxidants such as phenolics, ascorbic acid and flavonoids appears to be low. Need proper justification/evidences.

Additional comments

Need to be reoriented as the diseases identified is incorrect.

Reviewer 2 ·

Basic reporting

There are few suggestions regarding MS
1) The data on disease incidence to be required in table form for both seasons, if possible. as these data has already been collected and given in row data file.
2) The English language could possibly be improved.
3) The references in Line number 86, 93, 94, 110, 248,567,571,595, 735,870 are probably missing or year is not matched.

Experimental design

Experimental design is properly used and applied in research but It would be encourage if one more replication studies added by researchers to validate findings and enhance the overall reliability of the research.

Validity of the findings

All given data, graphs and images are well organized and enough information to validate the findings by researchers.

Reviewer 3 ·

Basic reporting

This manuscript describes the effects of three oligosaccharides oligo-chitosan, oligo-galacturonides and oligo-chitosan on cauliflower plants regarding, growth, seed yield, photosynthesis and defense agaist pathogens.
The manuscript contains too many colloquial or literary al expressions that should not be present in the description of experimental results, introduction or discussion. The. experimental design and results are correct and references are appropriate in introduction and discussion. The manuscript should be re-written and reviewed by a scientific fluent-English speaker.

Experimental design

As mentioned bfore experimental design is appropriate and results are relevant and meaninful.

Validity of the findings

The of results presented in this manuscript showed impact and novelty. The statistical analysis is correct. Conclusions are explained by results.

Additional comments

Line 23, change to largely unknown
Line 24, of cauliflower, delete BU and 1
Line 26, early season stole showed better growth parameters than mid-season parameters.
Line 39, explain SPAD
Line 40, increased levels of chlorophylls, ascorbic acid and flavonoids, and decreased levels of H2O2 and MDA IC50. Line 43, change "joined the applause" to scientific language
Line 45, that will be further cultivated in field conditions
Line 51, delete "aristocratic"
Line 53, delete "treasure trove"
Line 59, delete "undispute royalty of the vegetal kingdom" and change to scientific words
Line 65, delete "are embarked in a mission" and change to "are responsible for"
Line 67, delete "have been crowned" and change to "are responsible for"
Lione 76, delete "the reigning monark"
Line 77, delete "formidable"
Line 81, "delete "a clarion call"
Line 93, delete "to get ride of" and change
Line 106, delete "its magic touch"
Line 176, oligo-galacturonides
Line 205, defiine SPAD
Line 329, delete "was a bit better" and change to "was slightly better"
Line 337, delete "was amplified" and change to "was increased"
Line 379, Changes in phenological traits induced by oligosaccharides
Line 381, Here,
Line 399, Changes in seed yield induced by oligosaccharides
Line 433, Changes in plant defense induced by oligosaccharides
Line 442, Free radicals
Line 569, delete "sub-tropical haven"
Line 599, Experiments were performed

---

## Round 0.2 · accepted · Accept

Your article is accepted for publication.

Reviewer 2 ·

Basic reporting

All Suggestions are incorporated in MS

Experimental design

All necessary suggestions were made correctly.

Validity of the findings

All given data, graphs and images are well organized and enough information to validate the findings by researchers.

Additional comments

All suggestions were made correctly in reference and year sections.

·

Basic reporting

The revised manuscript is of sufficient quality for acceptance. I am confident that the authors have diligently addressed all previous comments, and I have no further remarks to add.

Experimental design

The revised manuscript is of sufficient quality for acceptance. I am confident that the authors have diligently addressed all previous comments, and I have no further remarks to add.

Validity of the findings

The revised manuscript is of sufficient quality for acceptance. I am confident that the authors have diligently addressed all previous comments, and I have no further remarks to add.

Additional comments

The revised manuscript is of sufficient quality for acceptance. I am confident that the authors have diligently addressed all previous comments, and I have no further remarks to add.